# SELF-SUPERVISED GRAPH-LEVEL REPRESENTATION LEARNING WITH LOCAL AND GLOBAL STRUCTURE

## ABSTRACT

This paper focuses on unsupervised/self-supervised whole-graph representation learning, which is critical in many tasks including drug and material discovery. Current methods can effectively model the local structure between different graph instances, but they fail to discover the global semantic structure of the entire dataset. In this work, we propose a unified framework called **Lo**cal-instance and **G**lobal-semantic Learning (*GraphLoG*) for self-supervised whole-graph representation learning. Specifically, besides preserving the local instance-level structure, GraphLoG leverages a nonparametric strategy to learn *hierarchical prototypes* of the data. These prototypes capture the semantic clusters in the latent space, and the number of prototypes can automatically adapt to different feature distributions. We evaluate GraphLoG by pre-training it on massive unlabeled graphs followed by fine-tuning on downstream tasks. Extensive experiments on both chemical and biological benchmark datasets demonstrate the effectiveness of our approach.

## 1 INTRODUCTION

Learning informative representations of whole graphs is a fundamental problem in a variety of domains and tasks, such as molecule properties prediction in drug and material discovery (Gilmer et al., 2017; Wu et al., 2018), protein function forecast in biological networks (Alvarez & Yan, 2012; Jiang et al., 2017), and predicting the properties of circuits in circuit design (Zhang et al., 2019). Recently, Graph Neural Networks (GNNs) have attracted a surge of interest and showed the effectiveness in learning graph representations. These methods are usually trained in a supervised fashion, which requires a large number of labeled data. Nevertheless, in many scientific domains, labeled data are very limited and expensive to obtain. Therefore, it is becoming increasingly important to learn the representations of graphs in an unsupervised or self-supervised fashion.

Self-supervised learning has recently achieved profound success for both natural language processing, *e.g.* GPT (Radford et al., 2018) and BERT (Devlin et al., 2019), and image understanding, *e.g.* MoCo (He et al., 2019) and SimCLR (Chen et al., 2020). However, how to effectively learn the representations of graphs in a self-supervised way is still an open problem. Intuitively, a desirable graph representation should be able to preserve the *local-instance structure*, so that similar graphs are embedded close to each other and dissimilar ones stay far apart. In addition, the representations of a set of graphs should also reflect the *global-semantic structure* of the data, so that the graphs with similar semantic properties are compactly embedded, which benefits various downstream tasks, *e.g.* graph classification or regression. Such structure can be sufficiently captured by semantic clusters (Caron et al., 2018; Ji et al., 2019), especially in a hierarchical fashion (Li et al., 2020).

There are some recent works that learn graph representation in a self-supervised manner, such as local-global mutual information maximization (Velickovic et al., 2019; Sun et al., 2019), structural-similarity/context prediction (Navarin et al., 2018; Hu et al., 2019; You et al., 2020) and contrastive multi-view learning (Hassani & Ahmadi, 2020). However, all these methods are capable of modeling only the local structure between different graph instances but fail to discover the global-semantic structure. To address this shortcoming, we are seeking for an approach that is sufficient to model both the local and global structure of a given set of graphs.

To attain this goal, we propose a **Lo**cal-instance and **G**lobal-semantic Learning (*GraphLoG*) framework for self-supervised graph representation learning. In specific, for preserving the local similarity

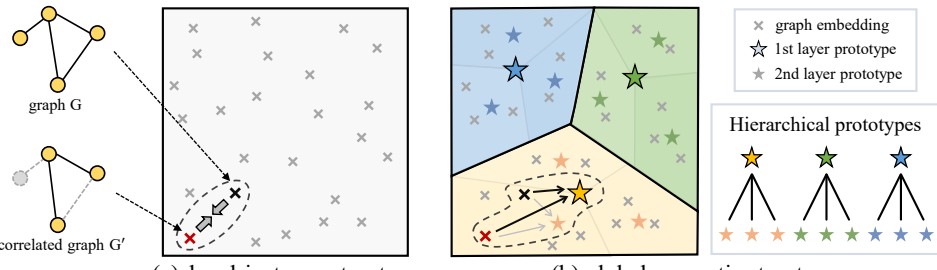

Figure 1: *Illustration of GraphLoG.* (a) Correlated graphs are constrained to be adjacently embedded to pursue the local-instance structure of the data. (b) Hierarchical prototypes are employed to discover and refine the global-semantic structure of the data.

between various graph instances, we first align the embeddings of correlated graphs/subgraphs[1] by maximizing their mutual information. In this locally smooth embedding space, we further represent the distribution of different graph embeddings with *hierarchical prototypes*[2] whose number is adaptively determined by the data in a nonparametric fashion. During training, these prototypes guide each graph to map to the semantically-similar feature cluster, and, simultaneously, the prototypes are maintained by online-updated graph embeddings. In this process, the global-semantic structure of the data is gradually discovered and refined. The whole model is pre-trained with a large number of unlabeled graphs, and then fine-tuned and evaluated on some downstream tasks.

We summarize our contributions as follows:

- We contribute a unified framework called Local-instance and Global-semantic Learning (GraphLoG) for self-supervised graph representation learning, which is able to model the structure of a set of graphs both locally and globally.

- We novelly propose to infer the global-semantic structure underlying the unlabeled graphs by learning hierarchical prototypes via a nonparametric strategy.

- We empirically verify our framework's superior performance on different GNN architectures through pre-training on a large-scale unlabeled dataset and fine-tuning on benchmark tasks in both the chemistry and biology domains.

## 2 Problem Definition and Preliminaries

### 2.1 Problem Definition

An ideal representation should preserve the local structure among the data instances. More specifically, we define it as follows:

**Definition 1 (Local-instance Structure).** The *local-instance structure* refers to the local pairwise similarity between different instances (Roweis & Saul, 2000; Belkin & Niyogi, 2002). To preserve the local-instance structure of graph-structured data, a pair of similar graphs/subgraphs, $\mathcal{G}$ and $\mathcal{G}'$, are expected to be mapped to the nearby positions of embedding space, as illustrated in Fig. 1(a), while the dissimilar pairs should be mapped to far apart.

The pursuit of local-instance structure is usually insufficient to capture the semantics underlying the entire dataset. It is therefore important to discover the global-semantic structure of the data, which is concretely defined as follows:

**Definition 2 (Global-semantic Structure).** A real-world dataset is usually distributed as different semantic clusters (Furnas et al., 2017; Ji et al., 2019). Therefore, we define the *global-semantic structure* of a dataset as the distribution of its semantic clusters, and each cluster is represented by a *prototype* (*i.e.* a representative cluster embedding). Since the semantics of a set of graphs can be structured in a hierarchical way (Ashburner et al., 2000; Chen et al., 2012), we represent the whole dataset with *hierarchical prototypes*. A detailed example can be seen in Fig. 1(b).

---

[1]In our method, we obtain correlated graphs/subgraphs via minor modification on node/edge attributes.

[2]Hierarchical prototypes are representative cluster embeddings organized in a hierarchical way.

**Problem Definition.** For *self-supervised graph representation learning*, a set of unlabeled graphs $\mathbb{G} = \{\mathcal{G}_1, \mathcal{G}_2, \cdots, \mathcal{G}_M\}$ are given, and we aim to learn a low-dimensional vector $h_{\mathcal{G}_i} \in \mathbb{R}^\delta$ for each graph $\mathcal{G}_i \in \mathbb{G}$ under the guidance of the data itself. In specific, we expect the derived graph embeddings $\mathbf{H} \in \mathbb{R}^{M \times \delta}$ follow both the local-instance and global-semantic structure.

## 2.2 PRELIMINARIES

**Graph Neural Networks (GNNs).** Given a graph $\mathcal{G} = (\mathcal{V}, \mathcal{E})$ with node attributes $X_\mathcal{V} = \{X_v | v \in \mathcal{V}\}$ and edge attributes $X_\mathcal{E} = \{X_{uv} | (u,v) \in \mathcal{E}\}$, a GNN aims to learn an embedding vector $h_v$ for each node $v \in \mathcal{V}$ and also a vector $h_\mathcal{G}$ for the entire graph $\mathcal{G}$. For an $L$-layer GNN, a neighborhood aggregation scheme is performed to capture the $L$-hop information surrounding each node. The $l$-th layer of a GNN can be formalized as follows:

$$h_v^{(l)} = \text{COMBINE}^{(l)}\left(h_v^{(l-1)}, \text{AGGREGATE}^{(l)}\left(\left\{\left(h_v^{(l-1)}, h_u^{(l-1)}, X_{uv}\right) : u \in \mathcal{N}(v)\right\}\right)\right), \quad (1)$$

where $\mathcal{N}(v)$ is the neighborhood set of $v$, $h_v^{(l)}$ denotes the representation of node $v$ at the $l$-th layer, and $h_v^{(0)}$ is initialized as the node attribute $X_v$. Since $h_v$ summarizes the information of a patch centered around node $v$, we will refer to $h_v$ as *patch embedding* to underscore this point. The entire graph's embedding can be derived by a permutation-invariant readout function:

$$h_\mathcal{G} = \text{READOUT}\left(\left\{h_v | v \in \mathcal{V}\right\}\right). \quad (2)$$

**Mutual Information Estimation.** Mutual information (MI) can measure both the linear and non-linear dependency between two random variables. Some recent works (Belghazi et al., 2018; Hjelm et al., 2019) employed neural networks to estimate the lower bound of MI. Among which, InfoNCE loss (van den Oord et al., 2018) has been introduced to maximize a lower bound of MI by minimizing itself, and we also adopt it in this work for its simplicity and effectiveness. In practice, given a query $q$, InfoNCE loss is optimized to score the positive sample $z_+$ higher than a set of distractors $\{z_i\}_{i=1}^K$:

$$\mathcal{L}_{\text{NCE}}\left(q, z_+, \{z_i\}_{i=1}^K\right) = -\log \frac{\exp\left(T(q, z_+)\right)}{\exp\left(T(q, z_+)\right) + \sum_{i=1}^K \exp\left(T(q, z_i)\right)}, \quad (3)$$

where $T(\cdot, \cdot)$ is a parameterized discriminator function which maps two representation vectors to a scalar value, whose architecture is detailed in Sec. 6.1.

**Rival Penalized Competitive Learning (RPCL).** The RPCL method (Xu et al., 1993) is a variant of classical competitive learning approaches, *e.g.* K-means clustering. Concretely, given a sample for update, RPCL-based clustering not only pulls the winning cluster center (*i.e.* the closest one) towards the sample, but also pushes the rival cluster center (*i.e.* the second closest one) away from the sample. We adopt this clustering algorithm for its strong capability of discovering feature clusters without specifying the number of clusters beforehand (*i.e.* in a nonparametric fashion), which is critical in the self-supervised learning scenarios where the number of semantic categories is not given.

## 3 LOCAL-INSTANCE AND GLOBAL-SEMANTIC LEARNING

### 3.1 LEARNING LOCAL-INSTANCE STRUCTURE OF GRAPH REPRESENTATIONS

We first define the *correlated graphs* that are expected to be embedded close to each other in the embedding space. Since the graphs from a dataset lie in a highly discrete space, it is hard to seek out the correlated counterpart of each graph from the dataset. To tackle this limitation, we propose to construct pairs of correlated graphs via the attribute masking strategy (Hu et al., 2019) which randomly masks a part of node/edge attributes in a graph (theoretical analysis is stated in Sec. A).

Through applying this technique to a randomly sampled mini-batch $B_\mathcal{G} = \{\mathcal{G}_j = (\mathcal{V}_j, \mathcal{E}_j)\}_{j=1}^N$ with $N$ graphs, the correlated counterpart of each graph can be obtained, which forms another mini-batch $B'_\mathcal{G} = \{\mathcal{G}'_j = (\mathcal{V}'_j, \mathcal{E}'_j)\}_{j=1}^N$ ($\mathcal{G}_j$ and $\mathcal{G}'_j$ are deemed as a pair of correlated graphs). Taking both mini-batches as input, the corresponding patch and graph embeddings are derived as follows:

$$h_{\mathcal{V}_j} = \{h_v | v \in \mathcal{V}_j\} = \text{GNN}(X_{\mathcal{V}_j}, X_{\mathcal{E}_j}), \quad h_{\mathcal{V}'_j} = \{h_v | v \in \mathcal{V}'_j\} = \text{GNN}(X_{\mathcal{V}'_j}, X_{\mathcal{E}'_j}), \quad (4)$$

$$h_{\mathcal{G}_j} = \text{READOUT}(h_{\mathcal{V}_j}), \quad h_{\mathcal{G}'_j} = \text{READOUT}(h_{\mathcal{V}'_j}), \quad (5)$$

where $h_{\mathcal{V}_j}$ ($h_{\mathcal{V}'_j}$) is the set of patch embeddings for graph $\mathcal{G}_j$ ($\mathcal{G}'_j$), and $h_{\mathcal{G}_j}$ ($h_{\mathcal{G}'_j}$) denotes the embedding of entire graph. With these ingredients, we design the learning objective for local-instance structure based on two desiderata: (1) similar subgraphs (*i.e.* patches) have similar feature representations; (2) graphs with a set of similar patches are embedded close to each other. To attain these goals, we propose to maximize the mutual information (*i.e.* minimize the InfoNCE loss) between correlated patches/graphs, which derives two constraints for the local-instance structure:

$$\mathcal{L}_{\text{patch}} = \frac{1}{\sum_{j=1}^N |\mathcal{V}'_j|} \sum_{j=1}^N \sum_{v' \in \mathcal{V}'_j} \sum_{v \in \mathcal{V}_j} \mathbb{1}_{v \leftrightarrow v'} \cdot \mathcal{L}_{\text{NCE}}\big(h_{v'}, h_v, \{h_{\tilde{v}} | \tilde{v} \in \mathcal{V}_j, \tilde{v} \neq v\}\big), \qquad (6)$$

$$\mathcal{L}_{\text{graph}} = \frac{1}{N} \sum_{j=1}^N \mathcal{L}_{\text{NCE}}\big(h_{\mathcal{G}'_j}, h_{\mathcal{G}_j}, \{h_{\mathcal{G}_k} | 1 \leqslant k \leqslant N, k \neq j\}\big), \qquad (7)$$

$$\mathcal{L}_{\text{local}} = \mathcal{L}_{\text{patch}} + \mathcal{L}_{\text{graph}}, \qquad (8)$$

where $\mathcal{L}_{\text{NCE}}(\cdot, \cdot, \cdot)$ is the InfoNCE loss function defined in Eq. 3, and $\mathbb{1}_{v \leftrightarrow v'}$ denotes the indicator function judging whether $v$ and $v'$ are the corresponding nodes in a pair of correlated graphs. Note that, masking node/edge attributes doesn't change the topology of a graph, which makes it easy to determine these corresponding nodes in our method.

## 3.2 LEARNING GLOBAL-SEMANTIC STRUCTURE OF GRAPH REPRESENTATIONS

It is worth noticing that the graphs in a dataset may possess hierarchical semantic information. For example, drugs (*i.e.* molecular graphs) are represented by a five-level hierarchy in the Anatomical Therapeutic Chemical (ATC) classification system (Chen et al., 2012). Moreover, the biological functions of proteins (*i.e.* graphs of amino acid residues) can be organized in a hierarchical structure (*e.g.* Gene Ontology (Ashburner et al., 2000) and FunCat (Ruepp et al., 2004) protein functional-definition schemes).

Motivated by this fact, we propose the notion of *hierarchical prototypes* to describe the distributions of graph embeddings. These prototypes are structured as a set of trees (Fig. 1(b)), in which each node denotes a prototype (*i.e.* a representative embedding of feature cluster) and corresponds to an unique parent node unless it is at the top layer. Formally, the hierarchical prototypes can be represented as $\{c_i^l\}_{i=1}^{M_l}$ ($l = 1, 2, \cdots, L_p$), where $L_p$ denotes the depth of hierarchical prototypes, and $M_l$ is the number of prototypes at the $l$-th layer. Except for the leaf nodes, each prototype possesses a set of child nodes, denoted as $\mathbb{C}(c_i^l)$ ($1 \leqslant i \leqslant M_l$, $l = 1, 2, \cdots, L_p - 1$). During training, the derivation of these variables can be divided into two stages, *i.e.* initialization and maintenance.

**Initialization of hierarchical prototypes.** In order to establish appropriate priors of graph embeddings, we first pre-train the GNN by minimizing $\mathcal{L}_{\text{local}}$ for one epoch and utilize it to extract the embeddings of all graphs in the training set, denoted as $\{h_{\mathcal{G}_i}\}_{i=1}^{N_D}$ ($N_D$ is the size of training set). These embeddings are used to initialize the bottom layer prototypes (*i.e.* $\{c_i^{L_p}\}_{i=1}^{M_{L_p}}$) via the RPCL-based clustering algorithm (Sec. 2.2):

$$\{c_i^{L_p}\}_{i=1}^{M_{L_p}} = \text{RPCL}\big(\{h_{\mathcal{G}_i}\}_{i=1}^{N_D}\big), \qquad (9)$$

where $\text{RPCL}(\cdot)$ outputs the cluster centers that are assigned with at least one sample. After that, the prototypes of upper layers are initialized by iteratively applying RPCL-based clustering to the prototypes of the layer below:

$$\{c_i^l\}_{i=1}^{M_l} = \text{RPCL}\big(\{c_i^{l+1}\}_{i=1}^{M_{l+1}}\big), \quad l = 1, 2, \cdots, L_p - 1. \qquad (10)$$

It is noteworthy that, in the initialization scheme, the number of prototypes is automatically adapted to the distribution of graph embeddings. As a result, this scheme is nonparametric and can adapt to different datasets without the prior knowledges about them.

**Maintenance of hierarchical prototypes.** In the training process, since the embedding of each graph is dynamically changing, we propose a strategy to maintain hierarchical prototypes with on-line updated graph embeddings. Concretely, under the guidance of a similarity measurement (*e.g.* cosine similarity in our implementation), the graph embeddings extracted from mini-batch $B_{\mathcal{G}}$ are

---

**Algorithm 1** Training procedure of Local-instance and Global-semantic Learning (GraphLoG).

---

**Input:** Training set $D = \{\mathcal{G}_j\}_{j=1}^{N_D}$, the number of training iterations $N_T$, hierarchical prototypes' depth $L_p$ and exponential decay rate $\beta$.
**Output:** The pre-trained GNN.
Initialize hierarchical prototypes $\{c_i^l\}_{i=1}^{M_l}$ $(l = 1, 2, \cdots, L_p)$
**for** $t = 1$ **to** $N_T$ **do**
    $B_{\mathcal{G}} \leftarrow \text{RandomSample}(D)$               # Get a mini-batch of graphs
    $B'_{\mathcal{G}} \leftarrow \text{AttrMasking}(B_{\mathcal{G}})$              # Get the correlated graphs
    $h_{\mathcal{V}_j}, h_{\mathcal{V}'_j}, h_{\mathcal{G}_j}, h_{\mathcal{G}'_j} \leftarrow$ Eqs. (4, 5) $(j = 1, 2, \cdots, N)$    # Extract patch and graph embeddings
    $\mathcal{L}_{\text{local}}, \mathcal{L}_{\text{global}} \leftarrow$ Eqs. (8, 13)           # Compute losses
    $\theta_{\text{GNN}} \overset{+}{\leftarrow} -\nabla_{\theta_{\text{GNN}}}(\mathcal{L}_{\text{local}} + \mathcal{L}_{\text{global}})$      # Update GNN's parameters
    $\theta_T \overset{+}{\leftarrow} -\nabla_{\theta_T}(\mathcal{L}_{\text{local}} + \mathcal{L}_{\text{global}})$        # Update discriminator's parameters
    $\{c_i^l\}_{i=1}^{M_l} \leftarrow$ Eqs. (11, 12) $(l = 1, 2, \cdots, L_p)$     # Maintain hierarchical prototypes
**end for**

---

divided into $M_{L_p}$ groups according to their most similar bottom layer prototype, and the mean graph embeddings are computed within each group, denoted as $\{\widehat{c}_i^{L_p}\}_{i=1}^{M_{L_p}}$. These mean embeddings are employed to update bottom layer prototypes via an exponential moving average scheme:

$$c_i^{L_p} \leftarrow \beta c_i^{L_p} + (1 - \beta)\widehat{c}_i^{L_p}, \quad 1 \leqslant i \leqslant M_{L_p}, \tag{11}$$

where $\beta$ is the exponential decay rate. For the prototypes of upper layers, they are updated with the mean of their child prototypes in the corresponding tree:

$$c_i^l \leftarrow \frac{1}{|\mathbb{C}(c_i^l)|} \sum_{c_k^{l+1} \in \mathbb{C}(c_i^l)} c_k^{l+1}, \quad 1 \leqslant i \leqslant M_l, \ l = 1, 2, \cdots, L_p - 1. \tag{12}$$

**Constraint for global-semantic structure.** Now that the latent semantic structure of the data has been represented by hierarchical prototypes, we seek to constrain the distributions of graph embeddings with these prototypes. The major goal here is to map correlated graphs to the same set of feature clusters. In practice, according to cosine similarity, we first search for the prototypes most similar to the embedding of graph $\mathcal{G}_j$ in each layer, denoted as $s(\mathcal{G}_j) = \{s_1(\mathcal{G}_j), s_2(\mathcal{G}_j), \cdots, s_{L_p}(\mathcal{G}_j)\}$. Note that, this search process follows the topology of hierarchical prototypes, which means that: $s_{l+1}(\mathcal{G}_j) \in \mathbb{C}(s_l(\mathcal{G}_j))$ $(l = 1, 2, \cdots, L_p - 1)$. Correspondingly, when using the embedding of the correlated graph $\mathcal{G}'_j$ for search, we expect an identical searching path, and such objective is pursued by maximizing the mutual information (*i.e.* minimizing the InfoNCE loss) between graph embedding $h_{\mathcal{G}'_j}$ and the prototypes in $s(\mathcal{G}_j)$:

$$\mathcal{L}_{\text{global}} = \frac{1}{N \cdot L_p} \sum_{j=1}^{N} \sum_{l=1}^{L_p} \mathcal{L}_{\text{NCE}}\big(h_{\mathcal{G}'_j}, s_l(\mathcal{G}_j), \{c_i^l | 1 \leqslant i \leqslant M_l, c_i^l \neq s_l(\mathcal{G}_j)\}\big). \tag{13}$$

**Discussion.** A recent work (Li et al., 2020) employed hierarchical prototypes for visual representation learning. The semantic hierarchy established in that work is derived from multiple times of clustering with different numbers of clusters, which relies on heuristically selected cluster numbers and fails to model the relations between the prototypes from different hierarchies. In contrast, our method is free from pre-defined cluster numbers, and a set of relational trees are constructed to embody the hierarchical relations between different prototypes.

## 3.3 MODEL OPTIMIZATION

The training procedure of Local-instance and Global-semantic Learning (GraphLoG) is summarized in Algorithm 1. Along training, using the online-updated graph embeddings, the constraints for local-instance and global-semantic structure are derived, and the hierarchical prototypes are maintained. In each iteration, the parameters of GNN and discriminator are optimized with gradient descent using the following objective:

$$\min_{\text{GNN}, T} \mathcal{L}_{\text{local}} + \mathcal{L}_{\text{global}}. \tag{14}$$

# 4 SUP-GRAPHLOG: A SUPERVISED BASELINE FOR LOCAL-INSTANCE AND GLOBAL-SEMANTIC LEARNING

In order to verify the effectiveness of local-instance and global-semantic learning when it is directly applied to supervised downstream tasks, we propose a baseline model, named as sup-GraphLoG, which combines a plain GNN and the proposed hierarchical prototypes.

In the training phase, in order to establish appropriate local-instance structure of graph embeddings, the GNN is first pre-trained along with a linear classifier to perform graph classification on the training set. For the initialization of hierarchical prototypes, the number of bottom layer prototypes is set as the class number of the supervised task (*e.g.* $2K_T$ bottom layer prototypes for a task with $K_T$ binary classification problems), and each bottom layer prototype is the mean embedding of all the training graphs belonging to the corresponding class. The upper layer prototypes are initialized as in Eq. 10. For maintenance, given a mini-batch of labeled graphs, each bottom layer prototype is updated by the mean embedding of all the graphs belonging to the corresponding class using an exponential moving average scheme as in Eq. 11, and the prototypes of upper layers are maintained following Eq. 12.

For constraining the global-semantic structure, compared with the top-down search in the self-supervised model (Sec. 3.2), in this supervised setting, we first use the label of graph $\mathcal{G}_j$ to randomly select a matched bottom layer prototype $s_{L_p}(\mathcal{G}_j)$ and then obtain the whole searching path $s(\mathcal{G}_j) = \{s_1(\mathcal{G}_j), s_2(\mathcal{G}_j), \cdots, s_{L_p}(\mathcal{G}_j)\}$ from bottom to up. Based on this positive searching path, we randomly sample a negative path $s^n(\mathcal{G}_j) = \{s_1^n(\mathcal{G}_j), s_2^n(\mathcal{G}_j), \cdots, s_{L_p}^n(\mathcal{G}_j)\}$ satisfying that graph $\mathcal{G}_j$ does not belong to the corresponding class of $s_{L_p}^n(\mathcal{G}_j)$, and $s_l^n(\mathcal{G}_j) \neq s_l(\mathcal{G}_j)$, $s_{l+1}^n(\mathcal{G}_j) \in \mathbb{C}(s_l^n(\mathcal{G}_j))$ $(l = 1, 2, \cdots, L_p - 1)$. It is expected that the embedding of graph $\mathcal{G}_j$ is more similar with the prototypes on path $s(\mathcal{G}_j)$ than the ones on path $s^n(\mathcal{G}_j)$, which defines the loss constraint on a mini-batch as follows:

$$\mathcal{L}_{\text{global}}^{\text{sup}} = \frac{1}{N \cdot L_p} \sum_{j=1}^{N} \sum_{l=1}^{L_p} \mathcal{L}_{\text{NCE}}\big(h_{\mathcal{G}_j}, s_l(\mathcal{G}_j), s_l^n(\mathcal{G}_j)\big). \tag{15}$$

We further optimize the GNN by minimizing this loss, which refines the global-semantic structure in the embedding space.

In the inference phase, given an unlabeled graph, we first compute the similarity between its embedding and all the bottom layer prototypes via the cosine similarity function. After that, the task-specific prediction is derived by comparing the similarity scores of the classes corresponding to these prototypes, in which the classes with larger scores serve as the prediction result.

# 5 RELATED WORK

**Graph Neural Networks (GNNs).** Recently, following the efforts of learning graph representations via optimizing random walk (Perozzi et al., 2014; Tang et al., 2015; Grover & Leskovec, 2016; Narayanan et al., 2017) or matrix factorization (Cao et al., 2015; Wang et al., 2016) objectives, GNNs are proposed to explicitly derive proximity-preserved feature vectors in a neighborhood aggregation manner. As suggested in Gilmer et al. (2017), the forward pass of most GNNs can be depicted in two phases, *Message Passing* and *Readout* phase, and various works (Kipf & Welling, 2017; Hamilton et al., 2017; Velickovic et al., 2018; Ying et al., 2018; Zhang et al., 2018; Xu et al., 2019) sought to improve the effectiveness of these two phases. Unlike these methods which are mainly trained in a supervised fashion, our approach aims for unsupervised/self-supervised learning for GNNs.

**Self-supervised Learning for GNNs.** There are some recent works that explored self-supervised graph representation learning with GNNs. García-Durán & Niepert (2017) learned graph representations by embedding propagation, and Velickovic et al. (2019), Sun et al. (2019) and Hassani & Ahmadi (2020) achieved this goal through mutual information maximization. Also, some self-supervised tasks, *e.g.* edge prediction (Kipf & Welling, 2016), context prediction (Hu et al., 2019; Rong et al., 2020a), graph partitioning (You et al., 2020) and edge/attribute generation (Hu et al., 2020), have been designed to acquire knowledges from unlabeled graphs. Nevertheless, all these methods are only able to model the local relations between different graph instances. The proposed framework seeks to discover both the local-instance and global-semantic structure of a set of graphs.

**Self-supervised Semantic Learning.** Clustering-based methods (Xie et al., 2016; Yang et al., 2016; 2017; Caron et al., 2018; Ji et al., 2019; Li et al., 2020) are commonly used to learn the semantic information of the data in a self-supervised fashion. Among which, DeepCluster (Caron et al., 2018) proved the strong transferability of the visual representations learnt by clustering prediction to various downstream visual tasks. Prototypical Contrastive Learning (Li et al., 2020) set a new state-of-the-art for unsupervised visual representation learning. These methods are mainly developed for images but not for graph-structured data. Furthermore, the hierarchical semantic structure of the data has been less explored in previous works.

# 6 EXPERIMENTS

## 6.1 EXPERIMENTAL SETUP

**Pre-training details.** Following Hu et al. (2019), we adopt a five-layer Graph Isomorphism Network (GIN) (Xu et al., 2019) with 300-dimensional hidden units and a mean pooling readout function for performance comparisons (Secs. 6.2 and 6.3). The discriminator for mutual information estimation is formalized as: $T(x_1, x_2) = g(f(x_1), f(x_2))$, where $f(\cdot)$ is a projection function fitted by two linear layers and a ReLU nonlinearity between them, and $g(\cdot, \cdot)$ is a similarity function (*e.g.* cosine similarity in our method). In all experiments, we use an Adam optimizer (Kingma & Ba, 2015) (learning rate: $1 \times 10^{-3}$, batch size: 512) to train the model for 20 epochs. Unless otherwise specified, the hierarchical prototypes' depth $L_p$ is set as 3, and the exponential decay rate $\beta$ is set as 0.95. For attribute masking, 30% node attributes in molecular graphs are masked, and 30% edge attributes in Protein-Protein Interaction (PPI) networks are masked. These hyperparameters are selected by the grid search on the validation sets of four downstream molecule datasets (*i.e.* BBBP, SIDER, ClinTox and BACE), and their sensitivity is analyzed in Secs. 6.4 and F.

**Fine-tuning details.** For fine-tuning on a downstream task, a linear classifier is appended on the top of pre-trained GNN, and an Adam optimizer (classifier's learning rate: $1 \times 10^{-3}$, GNN's learning rate: $1 \times 10^{-4}$, batch size: 32) is employed to train the model for 100 epochs. For sup-GraphLoG, the GNN is first trained along with a linear classifier for 50 epochs using an Adam optimizer (learning rate: $1 \times 10^{-3}$, batch size: 32), and it is then fine-tuned under the guidance of hierarchical prototypes by an Adam optimizer (learning rate: $1 \times 10^{-4}$, batch size: 32). All reported results are averaged over five independent runs under the same configuration. Our approach is implemented with PyTorch (Paszke et al., 2017), and the source code will be released for reproducibility.

**Performance comparison.** We compare the proposed method with existing self-supervised graph representation learning algorithms (*i.e.* EdgePred (Kipf & Welling, 2016), InfoGraph (Sun et al., 2019), AttrMasking (Hu et al., 2019), ContextPred (Hu et al., 2019) and GraphPartition (You et al., 2020)) to verify its effectiveness. Following the setting in Hu et al. (2019), after pre-training GNN models with self-supervised methods, a graph-level multi-task supervised pre-training is conducted to achieve more transferable graph representations, and the performance on downstream tasks is respectively evaluated before and after this graph-level supervised pre-training.

## 6.2 EXPERIMENTS ON CHEMISTRY DOMAIN

**Datasets.** For fair comparison, we use the same datasets as in Hu et al. (2019). In specific, a subset of ZINC15 database (Sterling & Irwin, 2015) with 2 million unlabeled molecules is employed for self-supervised pre-training, and a preprocessed ChEMBL dataset (Mayr et al., 2018) with 456K labeled molecules is used for graph-level supervised pre-training. Eight binary classification datasets contained in MoleculeNet (Wu et al., 2018) serve as downstream tasks.

**Results.** Tab. 1 reports the performance of proposed GraphLoG method compared with other works. Among all self-supervised learning strategies, our approach achieves the best performance on seven of eight tasks, and a 3% performance gain is obtained in terms of average ROC-AUC. After applying a subsequent graph-level supervised pre-training, our models' performance is further promoted. In particular, a 2.9% increase is observed on the SIDER dataset. Also, the comparison between two supervised methods without self-supervised pre-training is presented in the table, the proposed sup-GraphLoG outperforms the vanilla GIN model with random initialization, which demonstrates the benefit of learning global-semantic structure. The training curves of eight downstream tasks are provided in Sec. G.

Table 1: Test ROC-AUC (%) on molecular property prediction benchmarks.

| Methods | BBBP | Tox21 | ToxCast | SIDER | ClinTox | MUV | HIV | BACE | Avg |
|---|---|---|---|---|---|---|---|---|---|
| Random | $65.8 \pm 4.5$ | $\mathbf{74.0} \pm 0.8$ | $63.4 \pm 0.6$ | $57.3 \pm 1.6$ | $58.0 \pm 4.4$ | $71.8 \pm 2.5$ | $75.3 \pm 1.9$ | $70.1 \pm 5.4$ | 67.0 |
| sup-GraphLoG (ours) | $\mathbf{71.1} \pm 0.3$ | $72.9 \pm 0.2$ | $\mathbf{63.8} \pm 0.1$ | $\mathbf{61.4} \pm 0.6$ | $\mathbf{64.0} \pm 0.6$ | $\mathbf{72.5} \pm 1.0$ | $\mathbf{76.7} \pm 0.5$ | $\mathbf{76.5} \pm 1.1$ | $\mathbf{69.9}$ |
| EdgePred (2016) | $67.3 \pm 2.4$ | $76.0 \pm 0.6$ | $64.1 \pm 0.6$ | $60.4 \pm 0.7$ | $64.1 \pm 3.7$ | $74.1 \pm 2.1$ | $76.3 \pm 1.0$ | $79.9 \pm 0.9$ | 70.3 |
| InfoGraph (2019) | $68.2 \pm 0.7$ | $75.5 \pm 0.6$ | $63.1 \pm 0.3$ | $59.4 \pm 1.0$ | $70.5 \pm 1.8$ | $75.6 \pm 1.2$ | $77.6 \pm 0.4$ | $78.9 \pm 1.1$ | 71.1 |
| AttrMasking (2019) | $64.3 \pm 2.8$ | $\mathbf{76.7} \pm 0.4$ | $\mathbf{64.2} \pm 0.5$ | $61.0 \pm 0.7$ | $71.8 \pm 4.1$ | $74.7 \pm 1.4$ | $77.2 \pm 1.1$ | $79.3 \pm 1.6$ | 71.1 |
| ContextPred (2019) | $68.0 \pm 2.0$ | $75.7 \pm 0.7$ | $63.9 \pm 0.6$ | $60.9 \pm 0.6$ | $65.9 \pm 3.8$ | $75.8 \pm 1.7$ | $77.3 \pm 1.0$ | $79.6 \pm 1.2$ | 70.9 |
| GraphPartition (2020) | $70.3 \pm 0.7$ | $75.2 \pm 0.4$ | $63.2 \pm 0.3$ | $61.0 \pm 0.8$ | $64.2 \pm 0.5$ | $75.4 \pm 1.7$ | $77.1 \pm 0.7$ | $79.6 \pm 1.8$ | 70.8 |
| GraphLoG (ours) | $\mathbf{73.9} \pm 0.7$ | $76.2 \pm 0.2$ | $\mathbf{64.2} \pm 0.5$ | $\mathbf{61.7} \pm 1.2$ | $\mathbf{78.6} \pm 1.5$ | $\mathbf{76.4} \pm 1.0$ | $\mathbf{78.2} \pm 0.6$ | $\mathbf{83.3} \pm 1.4$ | $\mathbf{74.1}$ |
| Supervised | $68.3 \pm 0.7$ | $77.0 \pm 0.3$ | $64.4 \pm 0.4$ | $62.1 \pm 0.5$ | $57.2 \pm 2.5$ | $79.4 \pm 1.3$ | $74.4 \pm 1.2$ | $76.0 \pm 1.0$ | 70.0 |
| EdgePred* (2016) | $66.6 \pm 2.2$ | $78.3 \pm 0.3$ | $\mathbf{66.5} \pm 0.3$ | $63.3 \pm 0.9$ | $70.9 \pm 4.6$ | $78.5 \pm 2.4$ | $77.5 \pm 0.8$ | $79.1 \pm 3.7$ | 72.6 |
| InfoGraph* (2019) | $68.4 \pm 1.0$ | $77.6 \pm 0.7$ | $65.3 \pm 0.4$ | $62.5 \pm 0.7$ | $73.8 \pm 1.9$ | $79.3 \pm 1.6$ | $78.0 \pm 1.1$ | $82.4 \pm 1.3$ | 73.4 |
| AttrMasking* (2019) | $66.5 \pm 2.5$ | $77.9 \pm 0.4$ | $65.1 \pm 0.3$ | $63.9 \pm 0.9$ | $73.7 \pm 2.8$ | $81.2 \pm 1.9$ | $77.1 \pm 1.2$ | $80.3 \pm 0.9$ | 73.2 |
| ContextPred* (2019) | $68.7 \pm 1.3$ | $78.1 \pm 0.6$ | $65.7 \pm 0.6$ | $62.7 \pm 0.8$ | $72.6 \pm 1.5$ | $\mathbf{81.3} \pm 2.1$ | $79.9 \pm 0.7$ | $84.5 \pm 0.7$ | 74.2 |
| GraphPartition* (2020) | $71.1 \pm 0.5$ | $77.4 \pm 0.4$ | $64.2 \pm 0.1$ | $63.4 \pm 0.2$ | $72.9 \pm 0.4$ | $78.2 \pm 0.7$ | $78.6 \pm 0.4$ | $80.4 \pm 0.2$ | 73.3 |
| GraphLoG* (ours) | $\mathbf{74.0} \pm 0.8$ | $\mathbf{78.5} \pm 0.2$ | $\mathbf{66.5} \pm 0.5$ | $\mathbf{64.6} \pm 0.8$ | $\mathbf{78.6} \pm 0.7$ | $79.5 \pm 1.3$ | $\mathbf{80.1} \pm 0.7$ | $\mathbf{85.1} \pm 0.9$ | $\mathbf{75.9}$ |

"*" denotes the model composed of a specific self-supervised pre-training and a subsequent graph-level supervised pre-training.

Table 2: Performance comparison and ablation study on biological function prediction benchmark.

(a) Test ROC-AUC (%) of different methods.

| Methods | ROC-AUC (%) |
|---|---|
| Random | $64.8 \pm 1.0$ |
| sup-GraphLoG (ours) | $\mathbf{67.6} \pm 0.8$ |
| EdgePred (Kipf & Welling, 2016) | $70.5 \pm 0.7$ |
| InfoGraph (Sun et al., 2019) | $70.7 \pm 0.5$ |
| AttrMasking (Hu et al., 2019) | $70.5 \pm 0.5$ |
| ContextPred (Hu et al., 2019) | $69.9 \pm 0.3$ |
| GraphPartition (You et al., 2020) | $71.0 \pm 0.2$ |
| GraphLoG (ours) | $\mathbf{72.8} \pm 0.4$ |

(b) Ablation study for different loss terms.

| $\mathcal{L}_{patch}$ | $\mathcal{L}_{graph}$ | $\mathcal{L}_{global}$ | ROC-AUC (%) |
|---|---|---|---|
| ✓ | | | $70.6 \pm 0.5$ |
| | ✓ | | $71.1 \pm 0.7$ |
| | | ✓ | $71.4 \pm 0.4$ |
| ✓ | ✓ | | $72.0 \pm 0.3$ |
| ✓ | | ✓ | $72.0 \pm 0.6$ |
| | ✓ | ✓ | $71.8 \pm 0.5$ |
| ✓ | ✓ | ✓ | $\mathbf{72.8} \pm 0.4$ |

## 6.3 EXPERIMENTS ON BIOLOGY DOMAIN

**Datasets.** For biology domain, following the settings in Hu et al. (2019), 395K unlabeled protein ego-networks are utilized for self-supervised pre-training, and the prediction of 5000 coarse-grained biological functions on 88K labeled protein ego-networks serves as graph-level supervised pre-training. The downstream task is to predict 40 fine-grained biological functions of 8 species.

**Results.** In Tab. 2a, we report the test ROC-AUC of various self-supervised learning techniques, and more results on biology domain can be found in Sec. C. It can be observed that the proposed GraphLoG method outperforms existing approaches with a clear margin, *i.e.* a $1.8\%$ performance gain. This result illustrates that the proposed scheme is beneficial to fine-grained downstream tasks. In addition, sup-GraphLoG is able to promote the performance of a plain GIN model by $2.8\%$ on this biological downstream task.

## 6.4 ANALYSIS

**Effect of different loss terms.** In Tab. 2b, we analyze the effect of three loss terms on biological function prediction, and we continue using the GIN depicted in Sec. 6.1 in this experiment. When each loss is independently applied (1st, 2nd and 3rd row), the loss for global-semantic structure performs best, which probably benefits from its exploration of data's semantic information. Through combining these losses, the full model (last row) achieves the best performance, which illustrates that the learning of local-instance and global-semantic structure are complementary to each other. We provide more ablation studies on different model components in Sec. E.

**Results on different GNNs.** Fig. 2(a) presents the effect of self-supervised pre-training on four kinds of GNNs, GCN (Kipf & Welling, 2017), GraphSAGE (Hamilton et al., 2017), GAT (Velick-ovic et al., 2018) and GIN (Xu et al., 2019). We can observe that the proposed GraphLoG scheme outperforms two existing methods, AttrMasking and ContextPred, on all configurations, and it avoids the performance decrease relative to random initialization baseline on GAT.

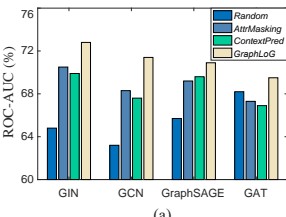 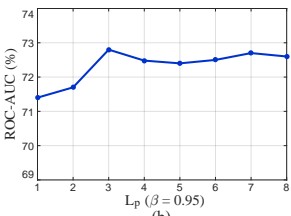 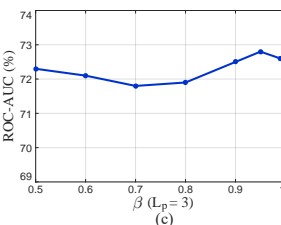

Figure 2: (a) Experimental results on different GNNs. (b)&(c) Sensitivity analysis of hierarchical prototypes' depth $L_p$ and exponential decay rate $\beta$. (All results are reported on biology domain.)

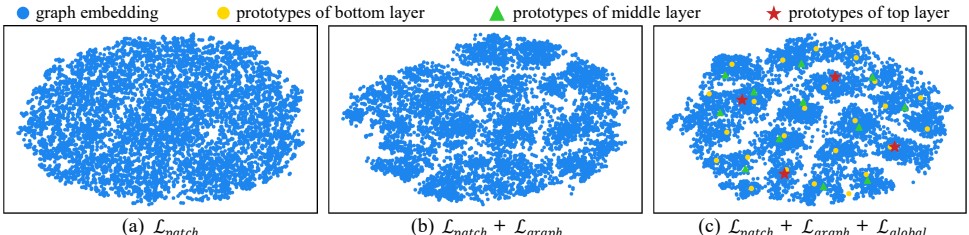

Figure 3: The t-SNE (Maaten & Hinton, 2008) visualization of graph embeddings and hierarchical prototypes on ZINC15 database (*i.e.* the pre-training dataset for chemistry domain).

**Sensitivity of hierarchical prototypes' depth $L_p$.** In this part, we discuss the selection of parameter $L_p$ which controls the number of discovered semantic hierarchies. In Fig. 2(b), we plot model's performance under different $L_p$ values. It can be observed that deeper hierarchical prototypes (*i.e.* $L_p \geqslant 3$) achieve stable performance gain compared to the shallow ones (*i.e.* $L_p \leqslant 2$).

**Sensitivity of exponential decay rate $\beta$.** In this experiment, we evaluate our approach's sensitivity to the parameter $\beta$. Fig. 2(c) shows the test ROC-AUC on downstream task using different $\beta$ values. From the line chart, we can observe that the proposed model's performance is not sensitive to $\beta$, which makes the maintenance scheme of hierarchical prototypes easy to tune.

**Visualization.** In Fig. 3, we utilize t-SNE (Maaten & Hinton, 2008) to visualize the distributions of graph embeddings and hierarchical prototypes on ZINC15 dataset. Compared to the model with only $\mathcal{L}_{patch}$ constraint, some feature clusters are formed after constraining the relations between correlated graphs' embeddings by $\mathcal{L}_{graph}$. More obvious feature separation is achieved after applying $\mathcal{L}_{global}$, which illustrates its effectiveness on discovering the global-semantic structure of the data.

## 7 CONCLUSIONS AND FUTURE WORK

We devise a unified framework called Local-instance and Global-semantic Learning (GraphLoG) for self-supervised graph representation learning, which models the structure of a set of unlabeled graphs both locally and globally. In this framework, we novelly propose to learn hierarchical prototypes upon graph embeddings to infer the global-semantic structure in graphs. Using the benchmark datasets from both chemistry and biology domains, we empirically verify our method's superior performance over state-of-the-art approaches on different GNN architectures.

Our future works will include exploring novel ways to construct correlated graphs, improving self-supervised learning manners, unifying pre-training and fine-tuning, and extending our framework to other domains such as sociology, physics and material science.

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

## A    THEORETICAL ANALYSIS OF CORRELATED GRAPH CONSTRUCTION

In our method, we choose the attribute masking (Hu et al., 2019) strategy to generate correlated graph pairs, which is widely used in recent self-supervised graph representation learning algorithms (Hu et al., 2019; 2020; Qiu et al., 2020). Since the graph structure has not been changed by the masking operation, the masked node attribute information can be partially recovered by its surrounding neighbors after being fed into a GNN. Therefore, the embeddings of correlated graph pairs can maintain a high degree of consistency in the feature space, which is desirable for the proposed GraphLoG model. We formally elucidate this point as follows.

Given an attributed graph $\mathcal{G} = (\mathcal{V}, \mathcal{E}, X_{\mathcal{V}}, X_{\mathcal{E}})$ ($X_{\mathcal{V}}$: node attributes, $X_{\mathcal{E}}$: edge attributes), we assume that its correlated graph $\mathcal{G}' = (\mathcal{V}, \mathcal{E}, X'_{\mathcal{V}}, X_{\mathcal{E}})$ is obtained by masking the attribute of a node $v$, i.e. $X'_{\mathcal{V}} \leftarrow X_{\mathcal{V}-\{v\}} \cup \{X_v^m\}$.

**Proposition 1.** *The L-layer GNN can repair the lost information induced by attribute masking operation by $\mathcal{I}_{\mathrm{repair}} \geqslant \mathcal{I}(X_v, \{X_{\tilde{v}} | \tilde{v} \in \mathcal{N}_v^L\})$, where $\mathcal{I}(\cdot, \cdot)$ denotes the mutual information, and $\mathcal{N}_v^L$ is the L-hop neighborhood set of $v$.*

*Proof.* Before information propagation by GNN, we define the lost information $\mathcal{I}_{\mathrm{lost}}$ induced by attribute masking as the conditional entropy of graph $\mathcal{G}$ conditioned on its correlated graph $\mathcal{G}'$:

$$\mathcal{I}_{\mathrm{lost}} = \mathcal{H}(\mathcal{G}|\mathcal{G}') = \mathcal{H}(X_v). \tag{16}$$

After the information propagation by GNN, we have:

$$\tilde{\mathcal{I}}_{\mathrm{lost}} = \mathcal{H}(h_{\mathcal{G}}|h_{\mathcal{G}'}) = \mathcal{H}(h_{\mathcal{G}}) - \mathcal{I}(h_{\mathcal{G}}, h_{\mathcal{G}'}), \tag{17}$$

where $\tilde{\mathcal{I}}_{\mathrm{lost}}$ is the information lost in the embedding of correlated graph $\mathcal{G}'$ compared with the embedding of origin graph $\mathcal{G}$.

According to the neighborhood aggregation scheme in GNN, we can derive:

$$\mathcal{I}(h_{\mathcal{G}}, h_{\mathcal{G}'}) = \mathcal{H}(h_{\mathcal{G}}) - \mathcal{H}(h_v) + \mathcal{I}(X_v, \{X_{\tilde{v}}|\tilde{v} \in \mathcal{N}_v^L\}), \tag{18}$$

where $\mathcal{I}(X_v, \{X_{\tilde{v}}|\tilde{v} \in \mathcal{N}_v^L\})$ denotes the recovered information for the masked node $v$ from its $L$-hop neighbors. Combining Eqs. 17 and 18 leads to:

$$\tilde{\mathcal{I}}_{\mathrm{lost}} = \mathcal{H}(h_v) - \mathcal{I}(X_v, \{X_{\tilde{v}}|\tilde{v} \in \mathcal{N}_v^L\}). \tag{19}$$

We can derive the information repaired by GNN, $\mathcal{I}_{\mathrm{repair}}$, as:

$$\mathcal{I}_{\mathrm{repair}} = \mathcal{I}_{\mathrm{lost}} - \tilde{\mathcal{I}}_{\mathrm{lost}} = \mathcal{I}(X_v, \{X_{\tilde{v}}|\tilde{v} \in \mathcal{N}_v^L\}) + (\mathcal{H}(X_v) - \mathcal{H}(h_v)). \tag{20}$$

Since $h_v$ is the low-dimensional embedding of node attribute $X_v$, we can deduce that:

$$\mathcal{H}(X_v) \geqslant \mathcal{H}(h_v). \tag{21}$$

Therefore, combining Eqs. 20 and 21, we can conclude:

$$\mathcal{I}_{\mathrm{repair}} \geqslant \mathcal{I}(X_v, \{X_{\tilde{v}}|\tilde{v} \in \mathcal{N}_v^L\}). \tag{22}$$

$\square$

We also evaluate the GraphLoG model with different correlated graph construction strategies, and the experimental results can be found in Sec. E.1. It empirically shows that attribute masking is more reliable to our method.

## B    MORE IMPLEMENTATION DETAILS

**Attribute masking strategy.** We add an extra dimension in the vector of node/edge attribute and set only that dimension as 1 when the corresponding node/edge is masked. Given a mini-batch of graphs, we mask the same proportion of node/edge attributes in each graph, and, for undirected graphs, the attributes on the both directions of an edge are masked/unmasked.

**GNN architecture.** All the GNNs in our experiments (*i.e.* GCN (Kipf & Welling, 2017), Graph-SAGE (Hamilton et al., 2017), GAT (Velickovic et al., 2018) and GIN (Xu et al., 2019)) are with 5 layers, 300-dimensional hidden units and a mean pooling readout function. In addition, two attention heads are employed in each layer of the GAT model.

## C More Results on Biology Domain

Table 3: Test ROC-AUC (%) on biological function prediction benchmark.

| Methods | ROC-AUC (%) | Methods | ROC-AUC (%) |
|---|---|---|---|
| Random | $64.8 \pm 1.0$ | Supervised | $72.9 \pm 0.5$ |
| EdgePred (Kipf & Welling, 2016) | $70.5 \pm 0.7$ | EdgePred* (Kipf & Welling, 2016) | $73.1 \pm 0.5$ |
| InfoGraph (Sun et al., 2019) | $70.7 \pm 0.5$ | InfoGraph* (Sun et al., 2019) | $73.7 \pm 0.4$ |
| AttrMasking (Hu et al., 2019) | $70.5 \pm 0.5$ | AttrMasking* (Hu et al., 2019) | $74.2 \pm 1.5$ |
| ContextPred (Hu et al., 2019) | $69.9 \pm 0.3$ | ContextPred* (Hu et al., 2019) | $74.3 \pm 0.6$ |
| GraphPartition (You et al., 2020) | $71.0 \pm 0.2$ | GraphPartition* (You et al., 2020) | $73.5 \pm 0.1$ |
| GraphLoG (ours) | $\mathbf{72.8} \pm 0.4$ | GraphLoG* (ours) | $\mathbf{75.7} \pm 0.6$ |

"*" denotes the model composed of a specific self-supervised pre-training and a subsequent graph-level supervised pre-training.

In Tab. 3, we report the performance of different approaches on the downstream task of biology domain, and the results before and after applying a subsequent graph-level supervised pre-training are respectively reported. It can be observed that the proposed GraphLoG method outperforms existing approaches with a clear margin under both settings, which illustrates the effectiveness of proposed learning scheme with and without the guidance of graph-level supervisory signal.

## D More Results on Graph Classification Benchmarks

Table 4: The 10-fold cross validation accuracy (mean $\pm$ std %) of self-supervised methods on graph classification benchmarks.

| Methods | MUTAG | PTC-MR | IMDB-Binary | IMDB-Multi | Reddit-Binary |
|---|---|---|---|---|---|
| random walk (Gärtner et al., 2003) | $83.7 \pm 1.5$ | $57.9 \pm 1.3$ | $50.7 \pm 0.3$ | $34.7 \pm 0.2$ | $-$ |
| node2vec (Grover & Leskovec, 2016) | $72.6 \pm 10.2$ | $58.6 \pm 8.0$ | $-$ | $-$ | $-$ |
| graph2vec (Narayanan et al., 2017) | $83.2 \pm 9.6$ | $60.2 \pm 6.9$ | $71.1 \pm 0.5$ | $50.4 \pm 0.9$ | $75.8 \pm 1.0$ |
| sub2vec (Adhikari et al., 2018) | $61.1 \pm 15.8$ | $60.0 \pm 6.4$ | $55.3 \pm 1.5$ | $36.7 \pm 0.8$ | $71.5 \pm 0.4$ |
| InfoGraph (Sun et al., 2019) | $89.0 \pm 1.1$ | $61.7 \pm 1.4$ | $73.0 \pm 0.9$ | $49.7 \pm 0.5$ | $82.5 \pm 1.4$ |
| Contrastive (Hassani & Ahmadi, 2020) | $89.7 \pm 1.1$ | $62.5 \pm 1.7$ | $74.2 \pm 0.7$ | $51.2 \pm 0.5$ | $84.5 \pm 0.6$ |
| GraphLoG (ours) | $\mathbf{89.9} \pm 1.5$ | $\mathbf{63.8} \pm 1.6$ | $\mathbf{76.6} \pm 4.2$ | $\mathbf{53.0} \pm 3.5$ | $\mathbf{85.9} \pm 2.9$ |

**Setups.** In this experiment, we compare GraphLoG with six self-supervised graph representation learning methods, *i.e.* random walk (Gärtner et al., 2003), node2vec (Grover & Leskovec, 2016), graph2vec (Narayanan et al., 2017), sub2vec (Adhikari et al., 2018), InfoGraph (Sun et al., 2019) and Contrastive (Hassani & Ahmadi, 2020). We strictly follow the linear evaluation protocol in Sun et al. (2019) and report the mean accuracy of 10-fold cross validation. Five conventional graph classification benchmark datasets, *i.e.* MUTAG (Kriege et al., 2016), PTC (Kriege et al., 2016), IMDB-Binary (Yanardag & Vishwanathan, 2015), IMDB-Multi (Yanardag & Vishwanathan, 2015) and Reddit-Binary (Yanardag & Vishwanathan, 2015), are used for evaluation. The settings of network architecture, optimizer and training parameters follow those in Sec. 6.1.

**Results.** Tab. 4 presents the comparisons of self-supervised approaches on five graph classification benchmark datasets. The proposed GraphLoG model ranks the first place in every task, and, especially, it outperforms a recent contrastive-learning-based method (Hassani & Ahmadi, 2020), which demonstrates the effectiveness of learning local-instance and global-semantic structure.

## E More Ablation Studies

### E.1 Ablation Study on Constructing Correlated Graphs

In this part, we analyze three ways of constructing correlated graphs, *i.e* AttrMasking (Hu et al., 2019) (with 30% attribute masking rate), DropEdge (Rong et al., 2020b) (with 10% edges dropped) and GraphDiffusion (Klicpera et al., 2019) (with heat kernel to derive a denser adjacency matrix),

Table 5: Ablation study for three model components on biological function prediction benchmark.

| Model Components | Methods | ROC-AUC (%) |
|---|---|---|
| Correlated Graph Construction | AttrMasking | **72.8** $\pm$ 0.4 |
| | DropEdge | 71.5 $\pm$ 0.5 |
| | GraphDiffusion | 70.4 $\pm$ 0.7 |
| Loss Format | InfoNCE | **72.8** $\pm$ 0.4 |
| | Hinge Loss | 72.3 $\pm$ 0.8 |
| Clustering Algorithm | K-means | 72.2 $\pm$ 0.3 |
| | RPCL | 72.8 $\pm$ 0.4 |
| | Adaptive-RPCL | **72.9** $\pm$ 0.3 |

and evaluate them under the proposed GraphLoG framework. As shown in the first segment of Tab. 5, the AttrMasking strategy outperforms other two techniques with a clear margin, which is mainly ascribed to the fact that, compared with dropping or adding edges, masking node attributes can preserve the matching degree of correlated graph pairs to a greater extent (referring to the theoretical analysis in Sec. A).

### E.2 ABLATION STUDY ON LOSS FORMAT

We investigate the effect of two loss formats, InfoNCE loss (van den Oord et al., 2018) and hinge-loss-based contrastive loss (Hadsell et al., 2006), on our model. The conventional contrastive loss uses one negative sample for each positive pair and is in a hinge loss form, while the InfoNCE loss employs a large number of negative samples and is in the form of softmax. We modify the loss format in Eqs. 6, 7 and 13 to conduct the comparison. According to the second segment of Tab. 5, the InfoNCE loss marginally improve model's performance, and both losses can achieve superior performance under the GraphLoG framework.

### E.3 ABLATION STUDY ON CLUSTERING ALGORITHM

We employ three clustering algorithms to derive hierarchical prototypes in our method and evaluate their corresponding pre-trained models on downstream task. For K-means, in the prototype initialization stage, we perform clustering hierarchically as in Eqs. 9 and 10 and obtain three hierarchies of prototypes with fixed number $M_1 = 10$, $M_2 = 30$ and $M_3 = 100$ for each hierarchy, respectively. Also, we design an adaptive variant of RPCL (Xu et al., 1993), named as Adaptive-RPCL, which is able to adjust the number of prototypes during training. Specifically, a counter is additionally maintained for each bottom layer prototype to record the number of iterations from the last time that the prototype is updated by graph embeddings. When a counter reaches threshold $\gamma = 100$, the corresponding bottom layer prototype is removed, and an upper layer prototype is removed if all the bottom layer prototypes in its corresponding tree are eliminated.

In the third segment of Tab. 5, the performance on biological downstream task is reported for the pre-trained models using different clustering algorithms. It can be observed that two RPCL-based clustering methods marginally outperform K-means, and their performance is comparable with each other. These results illustrate that the proposed GraphLoG model is not too sensitive to the selection of clustering algorithm.

### E.4 ROBUSTNESS OF CLUSTERING ALGORITHM

In this experiment, we examine the robustness of RPCL clustering algorithm in the proposed GraphLoG model. Specifically, we conduct clustering-based hierarchical prototype initialization for six times and obtain six different self-supervised pre-training models based on distinct initialization. Fig. 4 plots the biological downstream task performance of these six models. We can observe that they perform comparably with each other, which demonstrates that the GraphLoG model is fairly robust to different clustering outputs.

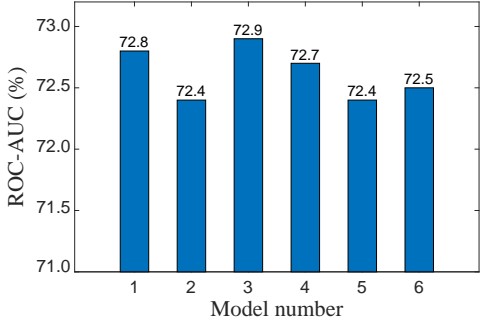

Figure 4: Test ROC-AUC (%) of six models derived by different clustering results on biological downstream task.

## F SENSITIVITY ANALYSIS OF ATTRIBUTE MASKING

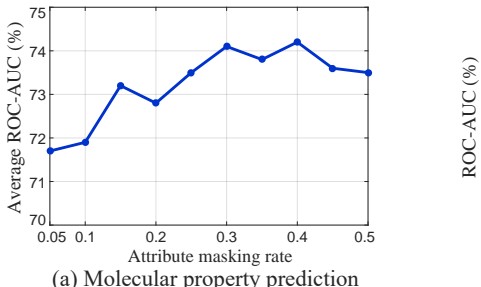
(a) Molecular property prediction

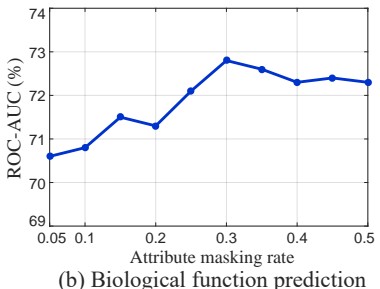
(b) Biological function prediction

Figure 5: Sensitivity analysis of attribute masking rate on (a) molecular property prediction benchmarks, where the average test ROC-AUC (%) on eight downstream tasks is reported, and (b) biological function prediction benchmark.

**Setups.** When varying the attribute masking rate to evaluate its sensitivity, other hyperparameters are fixed as the values depicted in Sec. 6.1. In specific, the hierarchical prototypes' depth $L_p$ is set as 3, and the exponential decay rate $\beta$ equals to 0.95.

**Results.** In Fig. 5, we plot model's performance on the downstream tasks of chemistry and biology domains under different masking rates. The highest test ROC-AUC is achieved when attribute masking rate is around 30%, which means that, under such settings, the constructed correlated graphs benefit the proposed learning scheme most.

## G TRAINING CURVES

In Fig. 6, we plot the training curves of four approaches, *i.e.* the random initialization baseline, context prediction (Hu et al., 2019), attribute masking (Hu et al., 2019) and the proposed GraphLoG method, on eight molecular property prediction tasks. From these line charts, we can observe that, through pre-training on a large-scale unlabeled dataset by GraphLoG, GNN model is able to converge at a higher ROC-AUC on the training set compared with other three methods.

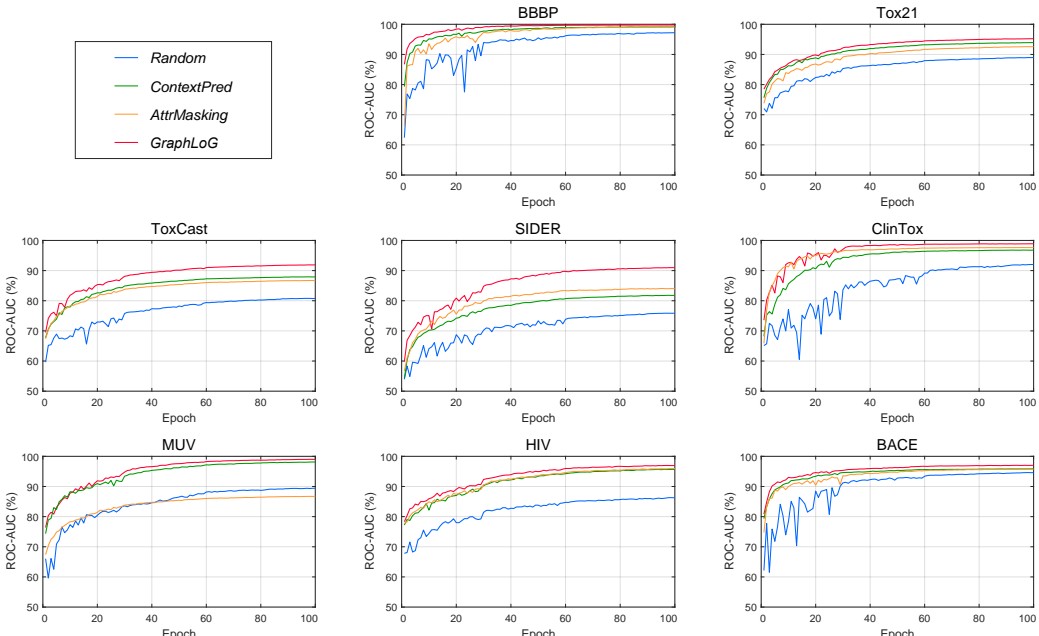

Figure 6: Training curves of different methods on eight downstream tasks of chemistry domain. The ROC-AUC (%) on training set is recorded along the training process.

