# OpenReview forum: "Self-supervised Graph-level Representation Learning with Local and Global Structure"
_ICLR.cc/2021/Conference — Reject_

### Official Review · AnonReviewer3 · 2020-10-26
**A combination of too many existing ideas that shadows the novelty and makes it hard to judge who should be given the credit of the empirical performance gains**

**Rating:** 5
**Confidence:** 4

**Review:**

This paper proposed a method for self-supervised graph-level representation learning. The main idea is to enforce both the instance level smoothness embedding constraints, and a so-called global, semantic grouping structures across all instance graphs in the training data set.  To achieve this goal, the authors have adopted a global clustering framework to encourage the embedding of the graphs belonging to the same clusters to be close to each other, and by using a hierarchically organized set of prototypes. The proposed method is applied to pre-train GNN on massive unlabeled graphs, which is then fine-tuned to downstream learning tasks.

Enforcing a global clustering structure can be useful in capturing the distribution of large number of graphs in the training data set and hopefully carry the learned representations over to other tasks. However, it appears to me that the paper has combined the carefully devised ideas from too many existing work, each of which alone has shown great success in improving the learning performance. Therefore it can be difficult to judge which part of the choices really leads to the final improvement, and in particular whether it is  the local and global structure preserving part, which seems to be the core theme of the paper (with the other theme being sel-supervised learning), that can fully explain the result.

In more detail, the authors have used (1) masking strategy by Hu et al., 2019 to generate correlated graph pairs, (2) mutual information estimation technique InfoNCE to enforce the correlation between paired graphs, and (3) Rival Penalized Competitive Learning (RPCL) as the main building block for hierarchical prototype-based learning. Therefore, a natural question to ask is, if one uses plain GNN architecture of each graph and plug it in a (hierarchical) clustering framework (without self-supervised learning and RPCL), whether similar improvements in the learning performance can still be obtained? If not, then the gains are merely due to the effectiveness of these specially designed components from the literatures and not by the general idea of local and global structures.

With regard to this concern, I would suggest the authors to clarify on what is the main theme of the work and demonstrate that the empirical performance gains are truly due to a novel, focused idea they propose rather than by combining some of the  existing algorithms which have shown great impact and performance gains in their respective context. In the current form, the novelrity of the work seems less significant by introducing so many components from other works.

---

> ### Author Response · Authors · 2020-11-17
> **Official Response to AnonReviewer3**
>
> Thanks for your insightful comments and great suggestions, which definitely help us improve the quality of this work.
>
> We respond to your concerns about the novelty of this work as follows:
>
> Q1: The combination of too many existing techniques shadows the novelty of this work.
>
> A1: Our core idea is to learn both the local-instance and global-semantic structure of a set of graphs in a self-supervised fashion. This problem, to the best of our knowledge, has not been explored by previous works, which makes the proposed GraphLoG model novel as a whole. Indeed, some existing techniques, i.e. attribute masking [a], InfoNCE loss [b] and RPCL clustering [c], have been adopted to construct the entire model. However, the additional ablation studies in the **Section E** of appendix show that these techniques (except attribute masking) can be replaced by their vanilla counterparts without too much hurt to model’s performance, which demonstrates the effectiveness of local-instance and global-semantic learning, i.e. the key idea of this work.
>
> Q2: If we combine a plain GNN with hierarchical prototypes, can this model achieve superior performance?
>
> A2: Following your suggestion, we further design a supervised variant of GraphLoG, named as sup-GraphLoG, in **Section 4**. The sup-GraphLoG model establishes hierarchical prototypes on top of a plain GNN and performs graph classification by measuring the similarity between graph embeddings and prototypes. This model outperforms the plain GNN on chemical and biological benchmark datasets (shown in **Tabs. 1 and 2**), which illustrates the effectiveness of global-semantic learning under the supervised setting.
>
> In addition, we would like to point out that the sup-GraphLoG model does not perform as well as the GraphLoG model, which demonstrates the necessity of self-supervised pre-training on massive unlabeled graphs.
>
>
> **In the revised paper, you can refer to the bolded sections above for the detailed contents related to your concerns.**
>
>
> [a] Hu, Weihua, et al. "Strategies for Pre-training Graph Neural Networks." ICLR, 2020.
>
> [b] Oord, Aaron van den, Yazhe Li, and Oriol Vinyals. "Representation learning with contrastive predictive coding." arXiv:1807.03748 (2018).
>
> [c] Xu, Lei, Adam Krzyzak, and Erkki Oja. "Rival penalized competitive learning for clustering analysis, RBF net, and curve detection." IEEE Transactions on Neural networks, 1993.

---

> > ### Comment · AnonReviewer3 · 2020-11-23
> > **the performance gains are still attributed to the combination of the three main building blocks from literature**
> >
> > The Table~5 in the appendix shows that the attribute masking and PPCL and InfoNCE are crucial components for the performance gains of the proposed method. When replaced by vanilla version, the performance drops. In other words, it is by combining these existing technqiues that final performance gains are achieved, which makes the novelty of the paper less significant.

---

> > > ### Author Response · Authors · 2020-11-24
> > > **Official Response to AnonReviewer3**
> > >
> > > Thanks for your feedback!
> > >
> > > We would like to clarify that the core idea of this work (local-instance and global-semantic learning) can achieve superior performance without the help of existing effective techniques:
> > >
> > > 1. First, compared with previous methods for self-supervised graph representation learning (e.g. Edge Prediction [a], InfoGraph [b], Context Prediction [c], etc.), the combination of the GraphLoG model with vanilla model components (i.e. hinge-loss-based contrastive loss, K-means clustering), denoted as **GraphLoG-vanilla**, can also achieve superior performance on the biological downstream task, as shown in Table A. This phenomenon illustrates that the GraphLoG model itself can obtain decent performance gain and surpass the state-of-the-art approaches.
> > >
> > > Table A: The performance comparison among different self-supervised methods on the biological function prediction benchmark.
> > >
> > > |Method|ROC-AUC (%)|
> > > |:----:|:----:|
> > > |EdgePred [a]|$70.5\pm0.7$|
> > > |InfoGraph [b]|$70.7\pm0.5$|
> > > |AttrMasking [c]|$70.5\pm0.5$|
> > > |ContextPred [c]|$69.9\pm0.3$|
> > > |GraphPartition [d]|$71.0\pm0.2$|
> > > |GraphLoG-vanilla|$\textbf{72.2}\pm0.4$|
> > >
> > > 2. Second, we apply the idea of local-instance and global-semantic learning to the supervised setting and use the vanilla model components (i.e. hinge-loss-based contrastive loss, K-means clustering), which is a plain version of the sup-GraphLoG model in **Section 4**, denoted as **sup-GraphLoG-vanilla**. This vanilla supervised model outperforms the Graph Isomorphism Network (GIN) [e] with a clear margin on the biological function prediction benchmark, as shown in Table B.
> > >
> > > Table B: The performance comparison between two supervised models on biological function prediction benchmark.
> > >
> > > |Method|ROC-AUC (%)|
> > > |:----:|:----:|
> > > |GIN [e]|$64.8\pm1.0$|
> > > |sup-GraphLoG-vanilla|$\textbf{66.9}\pm0.7$|
> > >
> > >
> > > [a] Kipf, Thomas N., and Max Welling. "Variational graph auto-encoders." arXiv:1611.07308 (2016).
> > >
> > > [b] Sun, Fan-Yun, et al. "Infograph: Unsupervised and semi-supervised graph-level representation learning via mutual information maximization." ICLR, 2020.
> > >
> > > [c] Hu, Weihua, et al. "Strategies for Pre-training Graph Neural Networks." ICLR, 2020.
> > >
> > >
> > > [d] You, Yuning, et al. "When Does Self-Supervision Help Graph Convolutional Networks?." ICML, 2020.
> > >
> > > [e] Xu, Keyulu, et al. "How powerful are graph neural networks?." ICLR, 2019.

---

> > > > ### Comment · AnonReviewer3 · 2020-11-24
> > > > **on novelty of the work**
> > > >
> > > > This work has three main components:
> > > > (1) identify local views/parts of a graph,
> > > > (2) enfoce contrastive loss, and
> > > > (3) enforce a global clustering.
> > > >
> > > > Among them, (1) follows a masking idea already proposed; for (2), Regardless of the hinge loss or mutual information loss, both are known in the literature and the idea of contrastive representation learning is not new, and so not a contribution.
> > > >
> > > > Then the main novelty of the work is to add a component of global clustering on top of the contrastive loss. I feel that the novelty of the work is still a concern.

---

> > > > > ### Author Response · Authors · 2020-11-25
> > > > > **Official Response to AnonReviewer3**
> > > > >
> > > > > Thanks for your insightful feedback!
> > > > >
> > > > > Exactly as you said, the main contribution of this work lies in modeling the global-semantic structure of graph embeddings via clustering different graphs in a hierarchical fashion. To the best of our knowledge, it is the first attempt to explore the semantic structure of a set of graphs in an unsupervised/self-supervised way. In the proposed GraphLoG framework, this global clustering is established on the local instance-level structure which is an essential prior guarantee for the subsequent clustering.
> > > > >
> > > > > In summary, this work embodies its novelty mainly on the global-semantic learning part, and also possesses the contribution of unifying both the local-instance and global-semantic learning into a single framework.

---

> ### Comment · Area_Chair1 · 2020-11-23
> **Feedback necessary**
>
> Dear reviewer,
>
> The authors have responded to your comments below. Could you please go over the response and give feedback to the authors sometime soon? The interactive discussion deadline is this Tuesday and you will not be able to interact with the authors after the date.
>
> Thanks,
> AC

---

### Official Review · AnonReviewer4 · 2020-10-29
**An interesting work for self-supervised graph representation learning**

**Rating:** 8
**Confidence:** 4

**Review:**

Pros:
- The paper proposed a novel self-supervised learning method to embed graphs to vector space. Different from previous methods, the method proposed a global-semantic learning strategy to encourage the embeddings to form a hierarchical clustering structure.  Both the embedding network and the hierarchical structure can be jointly learned.

- Authors have provided extensive and convincing comparison results and numerical analysis to show the effectiveness of the method.

- The paper is well-organized and clearly written. To the best of my knowledge, the proposed method is technically feasible.

Cons:
- The number of prototypes is determined by RPCL and can not be adjusted in training.
- Clustering algorithms are usually not very robust. Since the prototypes of GraphLoG is initialized by RPCL, is the performance of GraphLoG robust?

---

> ### Author Response · Authors · 2020-11-17
> **Official Response to AnonReviewer4**
>
> Thanks very much for your recognition in our work!
>
> We address your two concerns as follows:
>
> Q1: The number of prototypes determined by RPCL cannot be adjusted during training.
>
> A1: In the **Section E.3** of appendix, we design an adaptive variant of RPCL clustering (Adaptive-RPCL) which is able to adjust the number of prototypes during training. Its performance on biological downstream task is comparable with that of vanilla RPCL clustering algorithm in our method, which shows that the proposed GraphLoG model is not too sensitive to the selection of clustering algorithm.
>
> Q2: Is the performance of GraphLoG robust to different clustering outputs?
>
> A2: In the **Section E.4** of appendix, we compare the performance of six pre-trained models derived by different clustering results. It is observed that the downstream task performance of these models is comparable with each other, which demonstrates that the GraphLoG model is fairly robust to different clustering outputs.
>
>
> **In the revised paper, you can refer to the bolded sections above for the detailed experimental results.**

---

> > ### Comment · AnonReviewer4 · 2020-11-25
> > **An interesting work for self-supervised graph representation learning**
> >
> > To my best knowledge, this is the first work that learns graph embedding by exploiting the clustering structure of graphs and proves its effectiveness by detailed experiments.
> >
> > -For Q2, does the result mean the initialization of the prototypes is not important? How about use random initialization?

---

> > > ### Author Response · Authors · 2020-11-25
> > > **Official Response to AnonReviewer4**
> > >
> > > Thanks for your appreciation of our work!
> > >
> > > We further respond to your question as follows:
> > >
> > > Q: Can randomly initialized prototypes also work in the proposed GraphLoG model?
> > >
> > > A: The results in **Section E.4** mainly illustrate that, when the hierarchical prototypes are initialized with different clustering initialization (i.e. different initial candidate cluster centers in the RPCL algorithm), the performance of the pre-trained model on downstream tasks is not affected so much.
> > >
> > > However, if we directly use the initial candidate cluster centers as the initial prototypes without conducting RPCL clustering, these prototypes cannot represent the local instance-level structure established by local-instance learning and thus derive random semantic structure that does not match with the graphs in the dataset, which will hurt the GraphLoG model’s ability of producing graph embeddings with meaningful semantic structure. Because of the limitation of time in the discussion period, the experimental verification of this point has not been done, and we will finish it as soon as possible.

---

> ### Comment · Area_Chair1 · 2020-11-23
> **Feedback necessary**
>
> Dear reviewer,
>
> The authors have responded to your comments below. Could you please go over the response and give feedback to the authors sometime soon? The interactive discussion deadline is this Tuesday and you will not be able to interact with the authors after the date.
>
> Thanks,
> AC

---

### Official Review · AnonReviewer1 · 2020-10-29
**This paper proposes GraphLoG for self-supervised graph-level representation learning. It can learn both local-instance and global-semantic information. Experiments are conducted on chemical and biological benchmark.**

**Rating:** 6
**Confidence:** 3

**Review:**

The motivation and novelty of the proposed method are good. However, the validation is kind of
weak.
I can understand that this papers follows the validation in Hu et al (2019), however, I feel that
two tasks (one on chemical benchmark and one on biological benchmark) may not be sufficient
to give a detailed idea of the improvement of the proposed GraphLoG over other baselines. I
think 3-5 tasks are much better.
For the ablation study in Section 5.4, these ablated items are good. However, I more would like
to see fluctuated parts in the proposed GraphLoG. One example may be: is there any different
choice/option for hierarchical prototype? Which one is good/bad? What is the reason. Or other
potential and similar examples exist in the proposed GraphLoG. I think this will help us to
understand GraphLoG more.

---

> ### Author Response · Authors · 2020-11-17
> **Official Response to AnonReviewer1**
>
> Thanks for your support to the motivation and methodology of this work!
>
> We address your concern on the experimental verification as follows:
>
> Q1: More benchmark tasks should be added to evaluate the proposed method.
>
> A1: We additionally evaluate the proposed model on five graph classification benchmarks in the **Section D** of appendix. These five benchmark datasets involve the classification on the molecular graphs and social networks, and they are commonly used in previous self-supervised graph representation learning literature [a,b].
>
> Q2: More ablation studies should be conducted in order to better understand the proposed GraphLoG model.
>
> A2: We conduct more thorough ablation studies on the correlated graph construction, loss constraint and clustering algorithm in the **Section E** of appendix.
>
>
> **In the revised paper, you can refer to the bolded sections above for the detailed experimental results.**
>
>
> [a] Sun, Fan-Yun, et al. "Infograph: Unsupervised and semi-supervised graph-level representation learning via mutual information maximization."  ICLR, 2020.
>
> [b] Hassani, Kaveh, and Amir Hosein Khasahmadi. "Contrastive Multi-View Representation Learning on Graphs." ICML, 2020.

---

> ### Comment · Area_Chair1 · 2020-11-23
> **Feedback necessary**
>
> Dear reviewer,
>
> The authors have responded to your comments below. Could you please go over the response and give feedback to the authors sometime soon? The interactive discussion deadline is this Tuesday and you will not be able to interact with the authors after the date.
>
> Thanks,
> AC

---

### Official Review · AnonReviewer2 · 2020-11-01
**Official Blind Review #2**

**Rating:** 5
**Confidence:** 5

**Review:**

This paper proposes an unsupervised framework to perform graph representation learning. The local-instance structure is learned by first gets patch-level and graph-level representations for each graph, then maximize the mutual information between both correlated patches and correlated graphs, which are decided by attribute masking strategy. The global-semantic structure is maintained by leveraging RPCL to derive hierarchical prototypes of the representation and maximizing the mutual information between correlated graph representation and the searching path in the prototypes.

Strengths:
+  This paper presents a framework to jointly consider the local instance structure and global-semantic structure of graphs. It is a meaningful direction and could be beneficial for explainability.
+  The experimental results are quite thorough with comparisons to several baseline methods. Moreover, the ablation study of different mechanisms is provided in the experiments.

Weaknesses:
-  The proposed model seems like a simple combination of several existing techniques and thus lacks novelty.
-  The performance of this model seems to heavily rely on the attribute masking strategy as all the operations are built upon the correlated graph pairing from the attribute masking strategy. But how reliable is this technique? It seems to be a bottleneck of the model, and I think there should be an explanation on this either theoretically or experimentally.

Overall, the proposes a reasonable model for learning hierarchical graph representations. However, the novelty is limited since the proposed method seems like a simple combination of several existing techniques.

Questions:

1.  As I mentioned earlier, I wonder how reliable the attribute masking strategy is. As graph matching is an extremely hard problem, can this strategy provide a reliable pairing between correlated graphs?
2.  It is not clear how to leverage prototypes in classification tasks? I understand that the prototypes serve to ensure a better structure of the embeddings, but when classifying graphs, I wonder whether embeddings and prototypes are both used or not?
3.  In the Constraint for the global-semantic structure part, the loss for a graph embedding includes both the representations for its correlated graph and its searching path consisting of several prototypes. When minimizing the loss, I wonder both representations of the correlated graph and prototypes are updated together? Or the prototypes are only updated in eq. (11) and (12).

---

> ### Author Response · Authors · 2020-11-17
> **Official Response to AnonReviewer2**
>
> Thanks for your insightful reviews on this work!
>
> We first would like to re-emphasize the novelty of this work. This paper is dedicated to self-supervised graph representation learning which preserves both the local-instance and global-semantic structure of a set of unlabeled graphs. The proposed GraphLoG model is novel as a whole, since this learning problem, to the best of our knowledge, has not been studied by previous works. During constructing the entire model, we adopt some existing techniques, i.e. attribute masking [a], InfoNCE loss [b] and RPCL clustering [c], to promote model’s performance. However, these techniques can be substituted with the vanilla counterpart, e.g. InfoNCE loss -> hinge-loss-based contrastive loss, RPCL -> K-means, without too much hurt to model’s effectiveness, which is analyzed by the additional ablation studies in the **Section E** of appendix. In summary, we utilize these techniques as the performance-boosting modules serving for the core idea, local-instance and global-semantic learning, instead of simply combining them together.
>
> We respond to your questions as follows:
>
> Q1: The theoretical and experimental analysis about the reliability of attribute masking strategy should be supplemented.
>
> A1: We give a theoretical analysis about GNN’s capability of repairing the information lost by attribute masking in the **Section A** of appendix, and also empirically show that the correlated graphs derived by attribute masking is more reliable in **Section E.1**.
>
> Q2: How to use hierarchical prototypes in graph classification tasks?
>
> A2: In the self-supervised model GraphLoG, since the hierarchical prototypes cannot directly correspond to the categories of downstream tasks, they are not employed during graph classification. To study this problem in-depth, we additionally design a supervised learning variant, sup-GraphLoG, in **Section 4** to verify the effectiveness of hierarchical prototypes on graph classification with explicit supervision. The sup-GraphLoG model outperforms the vanilla GNN on chemical and biological benchmark datasets (shown in **Tabs. 1 and 2**).
>
> Q3: In the global-semantic loss, are graph embedding and hierarchical prototypes optimized jointly?
>
> A3: Appreciate for this good question. In the current model, only the representation of correlated graph is updated by the global-semantic loss (Eq. 13), and hierarchical prototypes are only updated by Eqs. 11 and 12. The joint optimization of these two types of variables will be the direction of our future exploration.
>
>
> **In the revised paper, you can refer to the bolded sections above for the detailed contents related to your questions.**
>
>
> [a] Hu, Weihua, et al. "Strategies for Pre-training Graph Neural Networks." ICLR, 2020.
>
> [b] Oord, Aaron van den, Yazhe Li, and Oriol Vinyals. "Representation learning with contrastive predictive coding." arXiv:1807.03748 (2018).
>
> [c] Xu, Lei, Adam Krzyzak, and Erkki Oja. "Rival penalized competitive learning for clustering analysis, RBF net, and curve detection." IEEE Transactions on Neural networks, 1993.

---

> > ### Comment · AnonReviewer2 · 2020-11-24
> > **Feedback to the response**
> >
> > Thanks for the detailed response. My previous concerns are partially resolved.
> >
> > To employ prototypes for the classification task, the authors design a new supervised setting in which the number of bottom-layer prototypes is set as the number of classes in the dataset. This setting, however, may not be rigorous enough. Some datasets used in the experiments only contain two classes. If the number of bottom-layer prototypes is set as 2, how can the model form a hierarchical prototype structure? Even for the datasets with multiple classes, the number of classes may not be large enough to be used as the number of the bottom-layer prototypes.
> >
> > After reading the theoretical justification of the attribute masking, I realized that the correlated graphs are generated instead of choosing from the dataset. Then my previous concern about graph matching is resolved. However, I’m curious about whether minimizing the representation distance between each graph (patch) and its masked version is sufficient to form a good representation structure? In this way, it seems that only extremely similar graphs (with the same structure and only several node attribute differences) are pulled together while the distance between the majority of the graphs is not well handled. In real-world datasets like chemical molecules, graphs with the same structures may be very rare, so the distance between different graphs may not be well handled in this case.
> >
> > Besides, the given theoretical justification seems not related to the paper. It states that the repaired information is larger than the mutual information between the masked node and its L-hop neighbors, which implies that the amount of information that can be recovered depends on the amount of information within the masked node that is contained in its neighbors. It is necessary to justify why using attribute masking can ensure a good embedding structure (similar graphs being pulled together and dissimilar ones being pushed apart). It seems to be very difficult and would depend on the types of datasets being used.
> >
> > Overall, my concerns are partially resolved and I would like to raise my rating from 4 to 5.

---

> > > ### Author Response · Authors · 2020-11-24
> > > **Official Response to AnonReviewer2**
> > >
> > > Thanks for your insightful feedback and careful review of the revised paper!
> > >
> > > We further respond to your questions as follows:
> > >
> > > Q1: In many datasets, the number of categories is limited, which makes it hard to construct hierarchical prototypes with effective structure.
> > >
> > > A1: As you said, some datasets in MoleculeNet (i.e. BBBP, HIV and BACE) contain only one binary classification task and thus two categories for constructing bottom layer prototypes. Therefore, for implementing the sup-GraphLoG model on various datasets, we use different depths of hierarchical prototypes, as listed in Table A. We would like to point out that, in the proposed model, the complexity of such semantic hierarchy is determined by the tasks studied in various datasets, *i.e.* the structure of hierarchical prototypes is task-specific. As a result, although the semantic hierarchy is quite simple for some datasets (containing only one hierarchy), the hierarchical prototypes are able to refine the structure of graph embeddings according to the tasks studied in each dataset.
> > >
> > > Table A: The depth of hierarchical prototypes for different datasets in MoleculeNet.
> > >
> > > |Dataset|BBBP|Tox21|ToxCast|SIDER|ClinTox|MUV|HIV|BACE|
> > > |:----:|:----:|:----:|:----:|:----:|:----:|:----:|:----:|:----:|
> > > |Number of binary classification tasks|1|12|617|27|2|17|1|1|
> > > |Depth of hierarchical prototypes|1|2|4|3|2|2|1|1|
> > >
> > > Q2: How can the correlated graph pairs obtained by attribute masking effectively constrain the structure of graph embeddings?
> > >
> > > A2: We think the global-semantic loss (**Eq. (13)**) plays a critical role in constraining the global structure of graph embeddings. In specific, it can push the embeddings of the graphs with different structures but potentially similar semantics towards the same prototypes, such that the graph embeddings within a semantic cluster are embedded more compactly. Such statement is also demonstrated by the t-SNE visualization results on ZINC15 database (**Figure 3**), in which the embeddings of different molecules form more compact clusters after applying the global-semantic loss $\mathcal{L}_{global}$.

---

> ### Comment · Area_Chair1 · 2020-11-23
> **Feedback necessary**
>
> Dear reviewer,
>
> The authors have responded to your comments below. Could you please go over the response and give feedback to the authors sometime soon? The interactive discussion deadline is this Tuesday and you will not be able to interact with the authors after the date.
>
> Thanks,
> AC

---

### Comment · Area_Chair1 · 2020-11-20
**The end of the discussion phase approaching**

Dear Reviewers,

The authors have provided detailed responses to your comments. Could you please go over the responses from the reviewers, read the revision, and provide feedback since the authors can have interactions with you only by this Tuesday (24th)?. I sincerely thank you for your service in reviewing for ICLR.

Thanks, Area Chair

---

### Decision · Program_Chairs · 2021-01-07
**Final Decision**

**Decision:**

Reject

**Comment:**

This paper proposes a self-supervised learning method for learning representations for graph-structured data, with both local and global objectives. The local objective aims to maximize the mutual information between two correlated graphs generated with attribute masking [Hu et al. 19], with the InfoNCE loss [van den Oord et al. 18], and the global objective aims to cluster the graphs using the RPCL [Xu et al. 93] objective, which pulls the sample toward the closest cluster while pushing it away from the rival clusters. The proposed method is validated on standard graph classification benchmarks by training a linear classifier on top of the GNN pre-trained with it, and the results show that it largely outperforms existing graph pre-training methods.

This paper fell into a borderline case, receiving split reviews with two of the reviewers learning toward rejection, and two others proposing to accept. The reviewers in general agreed that the experimental validation is thorough (except for one reviewer), and some of the reviewers mentioned that the proposed idea of performing self-supervised learning at both local and global level makes sense. However, the negative reviewers were concerned with the limited novelty of the proposed method, since the proposed method seems like a simple combination of two objectives each of which are based on existing ideas (although the latter has not been explored for GNN pre-training). The reviewers had interactive discussions with the authors, and the authors provided detailed feedback. Yet, the reviewers were not convinced that the method has sufficient novelty to warrant publication even after the internal discussion period, and decided to keep their negative ratings.

I believe that this is a simple yet effective pre-training method for GNNs on graph-structured data. The proposed method of combining the local and global objective seems like a promising solution to learn a metric space that well-captures the graph-level similarity and also is well-separated for discriminative classification, and it may have some practical impact given its good performance on benchmark datasets. However, as the two negative reviewers mentioned, the paper in its current form is presented as a simple combination of existing approaches. The local objective is a slight modification of attribute masking strategy of [Hu et al. 19], and the global objective of clustering has been explored in self-supervised learning of CNNs for image data [Asano et al. 20]. Thus, I lean toward rejecting the paper, considering its relative novelty and quality.

However, I find the proposed work highly promising, and encourage the authors to further develop the method while also improving on the paper writing. I suggest the authors to focus more on the main idea of learning with both local and global objectives, without specifically tying each objective to any of the existing methods. The authors may consider various techniques for both local and global objectives (such as hinge loss-based contrastive loss with k-means clustering as shown in the response to R3), and suggest the proposed work as a more general framework.

[Asano et al. 20] Self-Labeling via Simultaneous Clustering and Representation Learning, ICLR 2020